# Numerical and Experimental Analysis of Lap Joints Made of Grade 2 Titanium and Grade 5 Titanium Alloy by Resistance Spot Welding

**DOI:** 10.3390/ma16052038

**Published:** 2023-03-01

**Authors:** Lacki Piotr, Niemiro-Maźniak Judyta

**Affiliations:** Faculty of Civil Engineering, Czestochowa University of Technology, 69 Dabrowskiego St., 42-201 Czestochowa, Poland

**Keywords:** RSW, lap joint, Ti grade 2, grade 5 titanium alloy

## Abstract

The paper presents the evaluation of the load capacity of lap joints and the distribution of plastic deformations. The influence of the number and arrangement of welds on the load capacity of the joints and the method of their failure was investigated. The joints were made using resistance spot welding technology (RSW). Two combinations of joined titanium sheets were analyzed: Grade 2–Grade 5 and Grade 5–Grade 5. Non-destructive and destructive tests were carried out in order to verify the correctness of the welds within the given parameters. All types of joints were subjected to a uniaxial tensile test on a tensile testing machine, using digital image correlation and tracking (DIC). The results of the experimental tests of the lap joints were compared with the results of a numerical analysis. The numerical analysis was performed using the ADINA System 9.7.2 and was based on the finite element method (FEM). The conducted tests showed that the initiation of cracks in the lap joints occurred in the place as the maximum plastic deformations. This was determined numerically and confirmed experimentally. The number of welds and their arrangement in the joint affected the load capacity of the joints. Depending on their arrangement, Gr2–Gr5 joints with two welds reached from approximately 149 to 152% of the load capacity of joints with a single weld. The load capacity of the Gr5–Gr5 joints with two welds ranged from approximately 176 to 180% of the load capacity of joints with a single weld. Observations of the microstructure of RSW welds in the joints did not show any defects or cracks. The microhardness test in the Gr2–Gr5 joint showed that the average hardness of the weld nugget decreased by approximately 10–23% when compared to a Grade 5 titanium alloy and increased by approximately 59–92% compared to Grade 2 titanium.

## 1. Introduction

Resistance Spot Welding (RSW) is a process used in metal lap joints. These are joints in which where the elements overlap. They are used in large numbers in aircraft constructions and the automotive industry. The RSW process is appreciated due to its high production efficiency, low operating cost and high degree of automation [1]. Resistance welding consists of joining two or more sheets as a result of heat generated by electrical resistance [2]. The joined elements are placed between water-cooled electrodes. At a high-intensity and at the right time of welding, an electric current is then passed through the joined elements [3]. At the joining place, the elements heat up to the melting point of the metal, forming a liquid weld nugget. The size of the weld nugget is determined by the welding time and the welding current intensity. After turning off the welding current, the molten metal solidifies, and a uniform joint is created. The weld nugget is cooled under the pressure of the electrodes. An alternative method of joining aluminum sheets is the refill friction stir spot welding (RFSSW) process [4]. This is a process that uses the frictional heating of metals in the solid state to join them. It consists of the local friction heating of the joint by a rotating tool. In the case of RSW, the joining process takes place in the liquid state of the metals. During resistance spot welding, significant changes occur in the mechanical and metallurgical properties of the welded areas and heat-affected zones. The study of these changes is very important for the strength of welded joints [5].

The main purpose of resistance spot welding is to achieve a strong joining. An important element of the process is the selection and optimization of the appropriate welding parameters. An analysis of existing work [6,7,8,9] shows that the proper selection of parameters increases the strength of the analyzed joints. In [7], experimental studies showed that increasing the electrode pressure, welding current and welding time increased the tensile strength of the welded samples. A similar conclusion regarding the effect of the current intensity on the strength of the weld was obtained in [6]. It has been found that the tensile shear load capacity of welded materials increases with an increase in the heat and the input associated with the welding current. This is due to the increase in the size in the weld nugget. Research [9] also showed that the size of the weld increases with an increasing current. This results in an increase in the tensile shear load capacity. However, further increasing the current after achieving optimal parameters had an adverse effect on the joints. In [8], studies were also carried out on the effect of the current intensity during resistance welding on the quality and load capacity of the RSW weld. A numerical analysis of joints with different weld diameters was also carried out. It was found that with the increase in the weld diameter, the load capacity of the joint increased and the plastic deformation decreased. However, applying too much current can cause molten metal spatter and reduce the joint strength.

The resistance welding process is used to join steel [9,10,11,12], aluminum alloys [13,14], magnesium alloys, copper alloys and to join materials with different mechanical properties [15] and titanium.

Titanium and its alloys are the materials of the future, thanks to their very good properties: a low density, ranging from 4.43 to 4.85 g/cm^3^ [16], and a high mechanical strength, ranging from Rm ≈ 240 MPa for CP1 to about 1750 MPa for the heat-treated β titanium alloys [17]. They are used in various industries. In addition, titanium is characterized by a high resistance to corrosion and good biocompatibility. It is used in aviation, automotive, construction, and medical engineering, e.g., for forming parts of surgical instruments [18]. Titanium is also used in drawpieces [16].

The wide application of titanium and its alloys in industry creates the need to develop research on joining methods. The analysis of current work shows that titanium and its alloys are also joined by means of electron beam welding (EBW) [19,20,21], laser beam welding (LBW) [22], gas tungsten arc welding (GTAW) [23] and friction stir welding (FSW). In recent years, an increasing amount of research is available on the resistance welding of lap joints of titanium sheets. Table 1 presents a list of works on resistance spot welding of titanium and titanium alloys.

In this paper, the own research, conducted in [27], was repeated and extended. The article [27] was presented to evaluate the bending load capacity of a composite beam made of titanium Grade 2 and titanium alloy Grade 5 using RSW technology. RSW welding parameters were selected on the basis of joint strength tests and a microstructure analysis. For a joint with a single weld, a preliminary numerical analysis was carried out using the ADINA system. In this work, the experimental tests were repeated, and then the influence of the arrangement and number of welds in the joint on its load capacity was verified using the numerical method. For this purpose, a thorough numerical analysis was carried in the ADINA program, which was based on the finite element method (FEM), for all joints made. Numerical models differing in the configuration and number of welds were made. Conditions were also used in the numerical analysis that accurately represented the behavior of the joint in the in the tensile shear test. In order to assess the impact of the RSW process on the analyzed joints and changes in the material properties, microhardness measurements and a detailed structural analysis were additionally performed.

The choice of Grade 2 titanium and a Grade 5 titanium alloy was dictated by their wide application in aircraft structures. The wide application possibilities of these materials generated the need to expand the research. Testing the load capacity of titanium joints, and especially the influence of the number and arrangement of welds in the joint, are therefore very important issues in the context of the work and strength of the whole structure.

## 2. Materials and Methods

In this paper, two combinations of joined sheets were investigated: Grade 2–Grade 5 (Gr2–Gr5) and Grade 5–Grade 5 (Gr5–Gr5). The designations of the analyzed materials according to various standards are presented in Table 2, and their chemical compositions are presented in Table 3.

Samples with a width of 25 mm and a length of 100 mm were cut from Grade 2 and Grade 5 titanium sheets with a thickness of 0.8 mm. Using the RSW technique, the samples were then joined with a Soudronic resistance welder. Three variants of joints were made, differing in the number and arrangement of welds: joints with a single weld (Figure 1a), joints with two welds arranged parallel to the direction of the tensile shear (Figure 1b) and joints with two welds arranged perpendicularly to the direction of tensile shear (Figure 1c).

Five samples for each type of joint were made. Their names were established according to the following scheme: Gr2–Gr5.1-5 and Gr5–Gr5.1-5. Joints Gr2–Gr5.1-3 and Gr5–Gr5.1-3 were subjected to strength analysis, while joints Gr2–Gr5.4-5 and Gr5–Gr5.4-5 were subjected to a metallographic analysis and a microhardness measurement. Welding parameters for all sample combinations and configurations are shown in Table 4.

In order to check the correctness of the welds with the shown parameters, external inspections and metallographic tests of the welds were carried out. The samples were cut perpendicularly to the sheet surface along the axis of the weld. They were then ground, polished and etched for metallographic tests. The etching solution consisted of 2 vol % HN0_3_, 2 vol % HF, and 96 vol % H_2_O. The microstructure was assessed using a Keyence VHX-7000 microscope (KEYENCE, Mechelen, Belgium). The microhardness measurement was performed on the Emco Test Durascan 70 G5 apparatus (EMCO-TEST Prüfmaschinen GmbH, Kuchl, Austria), using the Vickers method with a 0.05 kgf test load. The dwell time was 12 s.

The uniaxial tension test of all types of joints were performed on the Zwick Z050 testing machine (ZwickRoell GmbH & Co. KG, Ulm, Germany). The test speed was 2 mm/min. The course of the test was recorded using the digital image correlation (DIC) system, which is an optical 3D deformation measurement system. It allows for a non-contact, material-independent analysis of statically and dynamically loaded structures. It consists of two cameras, which record the change in the position of measurement points (created on the basis of a stochastic pattern applied to the surface of the samples), lamps that illuminate the examined area and a computer that stores and processes the recorded data.

The results of experimental studies of the lap joints were compared with the results of the numerical analysis performed in the ADINA System 9.7.2, program based on the finite element method. The material constants and the true stress—true strain curve introduced in the ADINA program were obtained in a static tensile test. The static tensile test was carried out for the sheets with a thickness of 0.8 mm, which were composed of technically pure Grade 2 titanium and a Grade 5 titanium alloy. For each of the materials, samples were cut at three angles to the rolling direction of the sheet: 0°, 45° and 90°, according to the diagram in Figure 2 and according to the standard [31].

In the numerical analysis, boundary conditions and the type and value of the load were assumed in accordance with the results of the experimental research.

For all joints with a single weld and for two welds arranged parallel and perpendicular to the direction of tensile shear, the numerical models were made of 27-node, 3D solid elements.

The total number of finite elements and the nodes of the models are presented in Table 5. The size, number of elements and the number of nodes in a weld are presented in Figure 3. The models had one 27-node element through the thickness.

Boundary conditions and displacement were applied to the surfaces shown in Figure 3. In order to obtain more similar results to the experimental tests, boundary conditions were also assumed on volumes that represented parts of the samples located in the jaws of the testing machine. The sheet elements were connected at the place of the welds at the nodes. Numerical models are shown in Figure 3.

On the basis of numerical analysis, maps of the plastic deformations of the tested joints were obtained. They were compared with maps of plastic deformations obtained in experimental tests during their registration with the DIC system.

## 3. Results

On the basis of experimental tests and for all configurations of the titanium joints, a force-displacement diagram from the tensile shear test, the method of sample failure and the distribution of plastic strains on the joint surfaces were obtained. Tension curves and strengthening curves for the tested materials were also obtained. Microstructures of welds and microhardness distributions are presented. On the basis of numerical analysis, distributions of plastic deformations on the surfaces of the analyzed joint models were obtained.

### 3.1. Results from the Tensile Shear Test of Titanium Joints

The results of strength tests for Gr2–Gr5 titanium joints are shown in Figure 4, and the results for the Gr5–Gr5 joints are shown in Figure 5. The force-displacement diagram shows the repeatability of results in the series for all types of titanium joints.

The results of the tensile shear test of the Gr2–Gr5 titanium joints were presented in Table 6.

Joints with a single weld Gr2–Gr5-1 obtained the lowest load capacity, demonstrating an average value of 5.68 kN. The highest load capacity was achieved by joints with two welds arranged parallel to the direction of tensile shear Gr2–Gr5-2, which had an average value of 8.65 kN.

The mechanism of failure of titanium joints Gr2–Gr5 with a single weld is shown in Figure 6a. In each sample, during the tensile shear test, welds were separated from one of the joined sheets. The welds did not fail. In the case of joints with two welds arranged parallel and perpendicular to the direction of tensile shear, the parent material, Grade 2, cracked outside the area of the welds (Figure 6b,c). An exception was the Gr2–Gr5-3.2 joint, which had welds separate from one of the joined sheets.

A summary of the results from the tensile shear test of the Gr5–Gr5 titanium joints is presented in Table 7.

The highest tensile shear load capacity was achieved by joints with two welds arranged parallel to the direction of tension shear Gr5–Gr5-2, i.e., an average force value of 13.75 kN. Joints with a single weld achieved the lowest load capacity, i.e., an average value of 7.63 kN. Differences in the values of the maximum forces transferred by all types of joints in the series did not exceed 4% in relation to the values of the average forces. Titanium joints Gr5–Gr5-1.1-3, with a single weld, were destroyed by tearing the weld from two joined sheets. In the all tested joints, the joining was not discontinued, as is shown in Figure 7a. The predominance of tensile stresses and bending stresses, causing the bending of the sheets, was observed here. In the case of the Gr5–Gr5-2.1-3 titanium joints, which had two welds arranged parallel to the tensile shear direction, a sheet crack occurred at the edge of one of the welds in each sample (Figure 7b). The welds did not fail. A similar fracture mechanism was observed at joints with two welds arranged perpendicular to the direction of the tension shear.

In each joint, there were cracks at the edges of both welds. In the Gr5–Gr5-3.2 sample, the joint was completely broken (Figure 7c). None of the welds failed.

### 3.2. Static Tensile Test Results

The tensile curves of pure titanium Grade 2 and titanium alloy Grade 5, depending on the sampling angle, are shown in Figure 8. Pure titanium Gr2 is characterized by a lower tensile strength and a higher plasticity than the Gr5 titanium alloy.

The hardening curves for the selected titanium samples are presented in Figure 9. The Gr5 titanium alloy obtained higher yield stress values than the technically pure Gr2 titanium. The hardening curves were approximated by the power function, using the equations presented in the diagram. Obtained curves were introduced to the Adina program and used in numerical calculations.

### 3.3. Microstructure of RSW Welds

The microstructures of welds in the titanium joints Gr2–Gr5 and Gr2–Gr5 are shown in Figure 10 and Figure 11. The RSW joint consists of three main zones: the weld nugget, HAZ (heat-affected zone) and the base material. On the prepared microstructures, a clear demarcation between the zones was visible due to the different features of the microstructure. The microstructure of the welds is differs significantly from the parent material. The grain of the weld nugget structure was coarse compared to the base material; this was due to the large amount of heat introduced and the slower cooling time of the weld. In the zone of the weld nugget, a coarse-grain acicular martensite phase was visible. In the Gr2–Gr5 joint, the transition zone was more pronounced in the Grade 2 material than in the Grade 5 material. In the shear tension test, in the case of joints with a single weld, the weld was isolated from the Grade 2 material. No cracks around the weld nugget were observed in either analyzed microstructure. Defects or pores in the weld were also not noticed.

### 3.4. Microhardness

Figure 12 shows the weld microhardness distributions for the Gr2–Gr5 joint. Measurements were made in two directions: horizontal and vertical. Two measurement lines were introduced for the horizontal direction. The first line (1) passed through the Grade 5 parent material, HAZ and weld nugget. The second line (2) passed through the Grade 2 parent material, HAZ and weld nugget. The third line (3) was perpendicular to the line 1 and 2 and passed through the Grade 5 and Grade 2 parent materials, HAZ and weld nugget. The positioning accuracy of the indenter was ±0.0035.

In addition, on the basis of the measurement points indicated in Figure 13, a microhardness contour map was made for the Gr2–Gr5 joint (Figure 14).

Figure 12 and Figure 14 confirm that the hardness of the Gr5 base material is higher than that of the Gr2 material. From approximately 300 to 350 HV0.05 were obtained. The average hardness of the weld nugget in relation to the parent material Gr5 was lower and averaged 270 HV0.05. The hardness of the Gr2 parent material was lower than the hardness of the weld nugget and ranged from approximately 140 to 170 HV0.05.

Figure 15 shows the horizontal and vertical weld microhardness distribution for the Gr5–Gr5 joint. The first line (1) passed through the Grade 5 parent material, HAZ and weld nugget. The second line (2) was perpendicular to the line 1 and also passed through the Grade 5 parent material, HAZ and weld nugget.

### 3.5. Numerical Analysis Results

The results of the numerical analysis of the plastic deformations for titanium joints Gr2–Gr5 for each weld arrangement configuration are shown in Figure 16, Figure 17 and Figure 18. The results have been scaled in order to more clearly illustrate the deformation of the joint during the tensile shear test. The analyzed joints had different distributions of plastic deformations on the obverse and reverse sides. Larger deformations were observed on the surfaces of the Gr2 sheets. The concentration of plastic strain was visible around the RSW weld or welds and in the parent material within the welds in all joints. Their maxima occurred on the inner surfaces of the sheets.

The results of the numerical analysis for the Gr5–Gr5 titanium joints are shown in Figure 19, Figure 20 and Figure 21. The distribution of plastic deformations was the same on both sides: obverse and reverse in all Gr5–Gr5 samples. The concentration of the plastic deformation was observed at the weld edge for single weld joints and for joints with two welds arranged parallel to the direction of the tensile shear. In the case of joints with two welds arranged perpendicularly to the direction of tensile shear, the largest deformation values were visible at the edges of both welds and in the parent material adjacent to the welds.

## 4. Discussion

Taking into account the direction of sampling from the sheet (0°, 45° and 90° to the direction of rolling), the results of the average tensile strength of the analyzed materials are shown in Figure 22a. Some variation in the tensile strength values was observed for samples of the same material that were taken from the sheet at different angles to the direction of rolling. This indicates the occurrence of anisotropy of the plastic properties of the sheets as a result of their cold rolling. The results of the average tensile shear strength for the analyzed titanium joints from Table 6 and Table 7. are summarized in Figure 22b.

Joints Gr2–Gr5-1 with a single weld reached about 75% of the average load capacity of the same joints for Gr5–Gr5-1. Joints with two welds arranged parallel and perpendicular to the direction of tensile shear for the individual materials obtained a similar tensile shear load capacity. The highest average load capacity among joints with two welds (regardless of the arrangement) was achieved by joints Gr5–Gr5. The Gr2–Gr5 joint with two welds in both configurations of the weld arrangement obtained approximately 63% of the load capacity of the same Gr5–Gr5 joints. The higher load capacity of all the Gr5–Gr5 joints than the Gr2–Gr5 joints (Figure 22b) was due to the higher tensile strength of the Grade 5 titanium alloy compared Grade 2 titanium, which is visible in Figure 22a. For the same reason, different methods of failure of the samples during the tensile shear test were observed. The welds in the Gr2–Gr5 joint during the tensile shear tests separated completely from one of the joined sheets—the Grade 2 sheet. In the case of the Gr5–Gr5 joints, the weld was torn off from both joined sheets but the joints were not broken. In the case of Gr2–Gr5 joints with two welds, the weaker Grade 2 material cracked outside of the area of the welds. In the Gr5–Gr5 joints with two welds, a crack occurred at the edge of one or both welds, depending on their arrangement.

Studies on the influence of the weld number and their arrangement in the joint showed that:-Gr2–Gr5.2 joints with two welds arranged parallel to the direction of tensile shear achieved approximately 152% of the average load capacity of Gr2–Gr5.1 joints with a single weld, whereas Gr2–Gr5.3 joints with two welds perpendicular to the direction of tensile shear achieved 149% of the average load capacity of these joints;-Gr5–Gr5.2 joints with two welds arranged parallel to the direction of tensile shear achieved approximately 180% of the average load capacity of Gr5–Gr5.1 joints with a single weld, whereas Gr2–Gr5.3 joints with two welds perpendicular to the direction of tensile shear achieved 176% of the average load capacity of these joints.

The microhardness distributions for the Gr2–Gr5 joint (Figure 10) and the microhardness contour map (Figure 14) showed that the average hardness of the weld nugget ranged from approximately 77 to 90% of the hardness of the Grade 5 titanium alloy and from approximately 159 to 192% of the hardness of the Grade 2 titanium. The hardness of the Grade 5 titanium alloy was more than twice as high as the hardness of the Grade 2 titanium; therefore, an increase in the hardness of the weld nugget was observed in relation to the Grade 2 parent material and a decrease in hardness was observed in relation to the Grade 5 parent material.

Figure 23, Figure 24, Figure 25 and Figure 26 present the results of experimental studies achieved using the DIC system and the results of numerical analyses performed in the Adina program for all configurations of the Gr2–Gr5 and Gr5–Gr5 joints.

In all the Gr2–Gr5 and Gr5–Gr5 joints analyzed, similar distributions of plastic deformations were obtained using both test methods. The value of deformation obtained from the experimental tests corresponded to the values obtained in the numerical analysis.

The maxima of plastic deformation in all cases were visible at the edges of the welds and in the parent material by the welds located on the side of the fixed end of the sheet. In all cases, the minima were located at the free ends of the sheets. In the case of Gr5–Gr5 joints, an identical distribution of plastic deformations was observed on the obverse and reverse sides. There was a noticeable slight asymmetry in the experimental studies. This resulted from the initial imperfections from the process of making the RSW joints (the joints were made by hand). In places of maximum plastic deformation, cracks were initiated in analyzed lap joints.

## 5. Conclusions

Based on the experimental research and numerical analysis, the following conclusions were drawn:-Among the analyzed joints, which were made of 0.8 mm thick sheets, Grade5–Grade5 joints achieved a higher tensile shear strength than the Grade5–Grade2 joints. This was due to the higher tensile strength of the Grade 5 titanium alloy than Grade 2 titanium;-Tests of joints made using RSW (resistance spot welding) showed that the load capacity of the joints was affected by the number of welds and their arrangement in the joint. Gr2–Gr5 joints with two welds, depending on their arrangement, achieved approximately 149 to 152% of the load capacity of joints with a single weld. The load capacity of Gr5–Gr5 joints with two welds ranged from approximately 176 to 180% of the load capacity of joints with a single weld;-Crack initiation in lap joints occurred in the place of occurrence of maximum plastic deformations. This was determined numerically and confirmed experimentally;-Observations of the microstructure of RSW joints made of the analyzed materials showed no defects and cracks;-The microhardness test of the Gr2–Gr5 joint showed that the average hardness of the weld nugget decreased by approximately 10–23% compared to the Grade 5 titanium alloy base material and increased by approximately 59–92% compared to the Grade 2 titanium base material. The observed decrease and increase in the hardness of the weld nugget in relation to its parent materials was caused by the difference in the hardness between the Grade 2 titanium and Grade 5 titanium alloy. The Grade 5 titanium alloy had more than twice the hardness of Grade 2 titanium.

## Figures and Tables

**Figure 1 materials-16-02038-f001:**
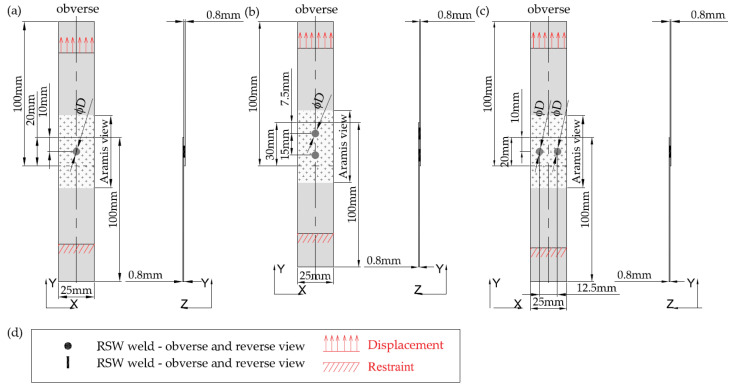
Geometry of the analyzed joints, mm: (**a**) joints with a single weld; (**b**) joints with two welds—parallel to the direction of tensile shear; (**c**) joints with two welds—perpendicular to the direction of tensile shear; and (**d**) legend.

**Figure 2 materials-16-02038-f002:**
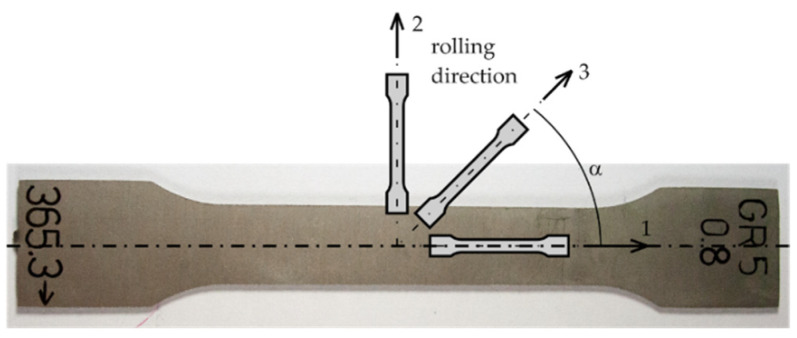
Scheme of taking samples cut from a sheet at different angles to the rolling direction: 0° (1), 45° (3) and 90° (2).

**Figure 3 materials-16-02038-f003:**
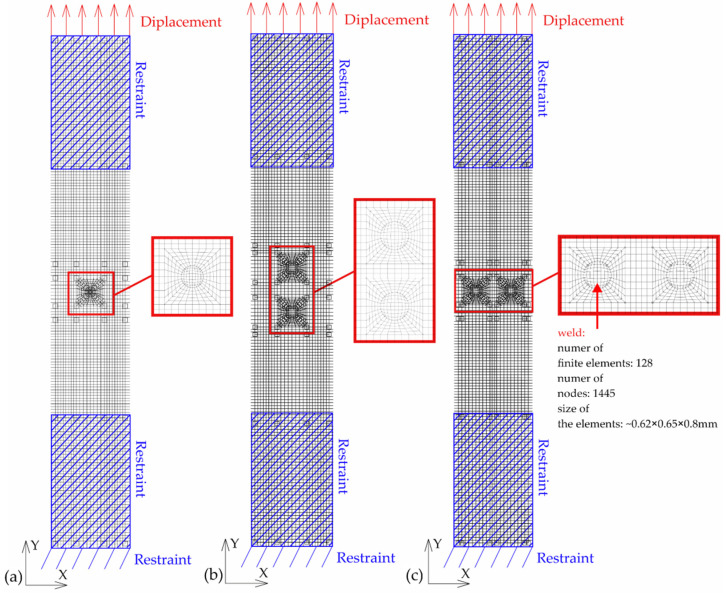
Numerical model with FEM mesh: (**a**) joints with single weld; (**b**) joints with two welds—parallel to the direction of tensile shear and (**c**) joints with two welds—perpendicular to the direction of tensile–shear.

**Figure 4 materials-16-02038-f004:**
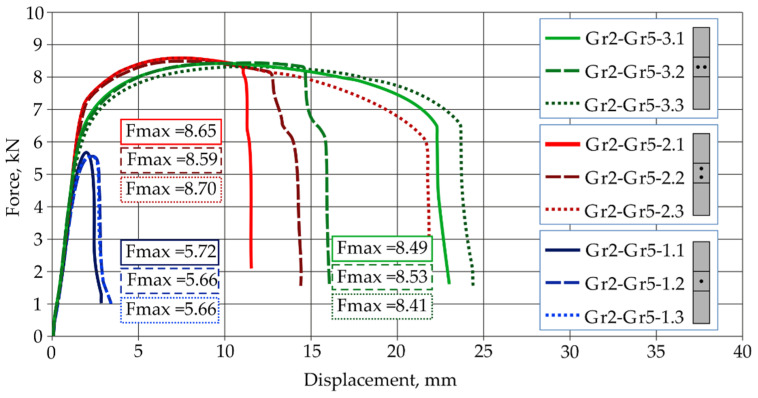
Force-displacement diagram from tensile shear test of Grade2–Grade 5 joints.

**Figure 5 materials-16-02038-f005:**
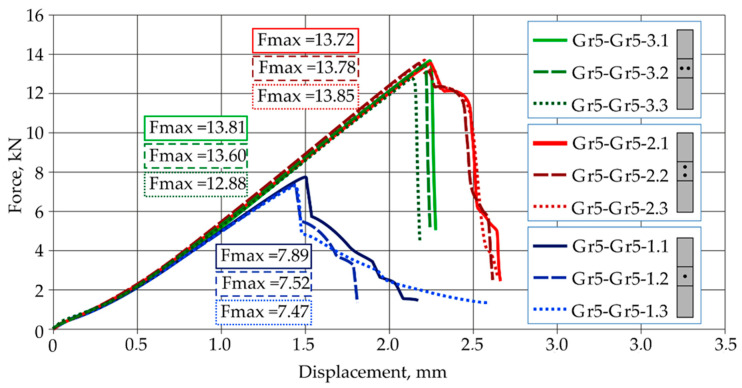
Force-displacement diagram from tensile shear test of Grade5–Grade 5 joints.

**Figure 6 materials-16-02038-f006:**
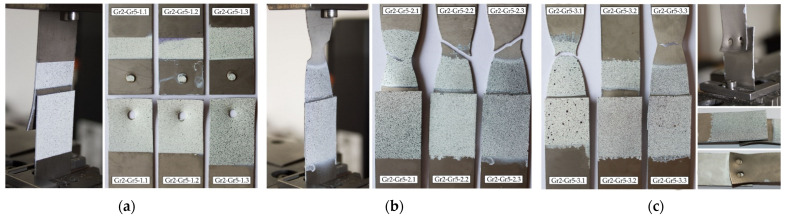
The method of failure of titanium samples in a tensile shear test: (**a**) with a single weld—Gr2–Gr5.1; (**b**) with two welds arranged parallel to the direction of tensile shear—Gr2–Gr5.2 and (**c**) with two welds arranged perpendicular to the direction of tensile shear—Gr2–Gr5.3.

**Figure 7 materials-16-02038-f007:**
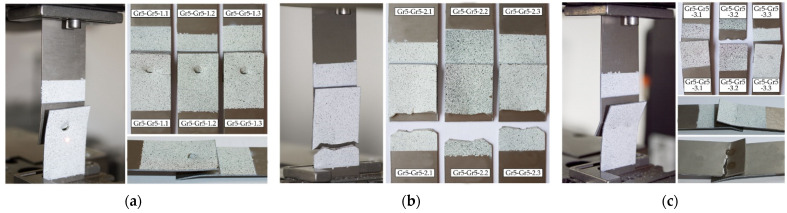
The method of failure of titanium samples in a tensile shear test: (**a**) with a single weld—Gr5–Gr5.1; (**b**) with two welds arranged parallel to the direction of tensile shear—Gr5–Gr5.2 and (**c**) with two welds arranged perpendicular to the direction of tensile shear—Gr5–Gr5.3.

**Figure 8 materials-16-02038-f008:**
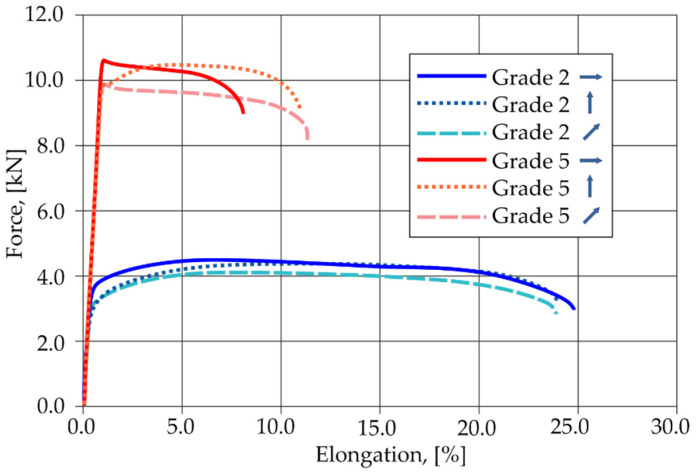
Tensile curves of Gr2 titanium and Gr5 titanium alloy, depending on the sampling angle.

**Figure 9 materials-16-02038-f009:**
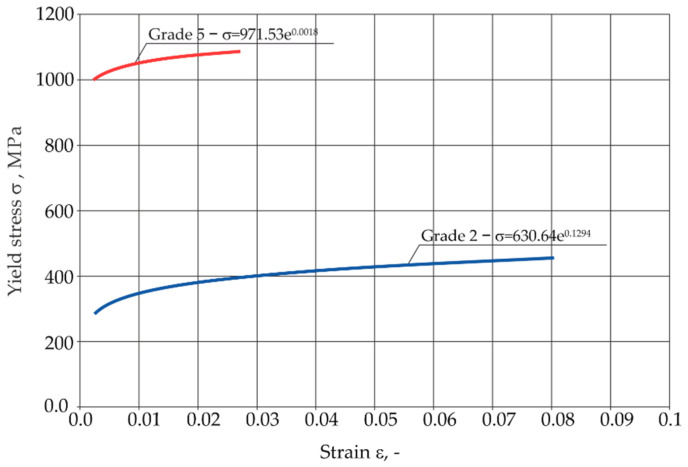
Hardening curves for the analyzed titanium samples.

**Figure 10 materials-16-02038-f010:**
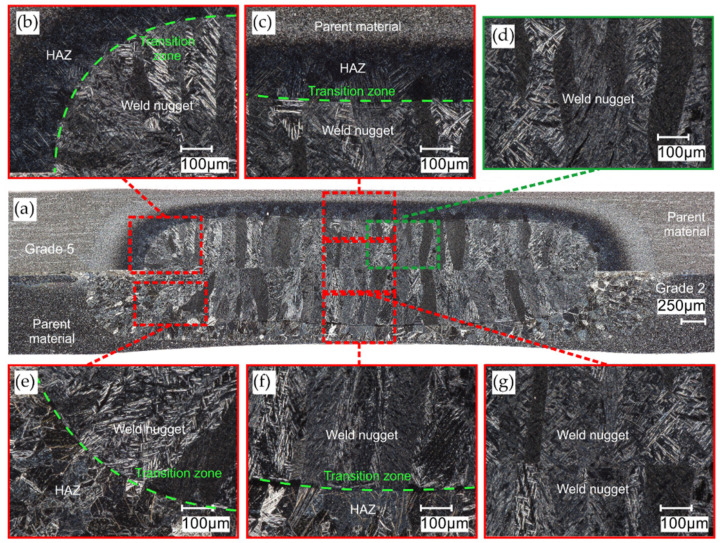
Microstructure of weld in the Gr2–Gr5 joint: (**a**) panorama of the weld; (**b**,**c**,**e**,**f**) transition zone between the weld nugget, HAZ and parent material; (**d**) weld nugget zone and (**g**) weld nugget center.

**Figure 11 materials-16-02038-f011:**
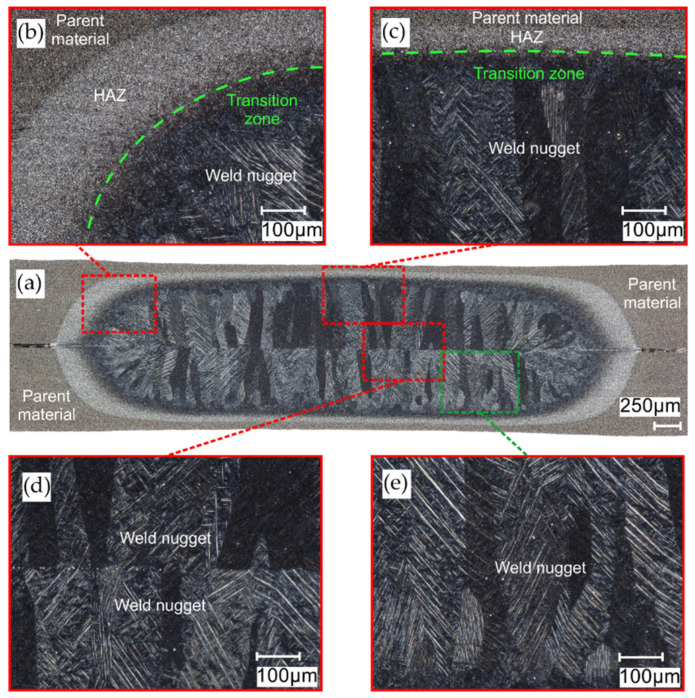
Microstructure of weld in the Gr5–Gr5 joint: (**a**) panorama of the weld; (**b**,**c**) transition zone between the weld nugget, HAZ and parent material; (**d**) weld nugget center and (**e**) weld nugget zone.

**Figure 12 materials-16-02038-f012:**
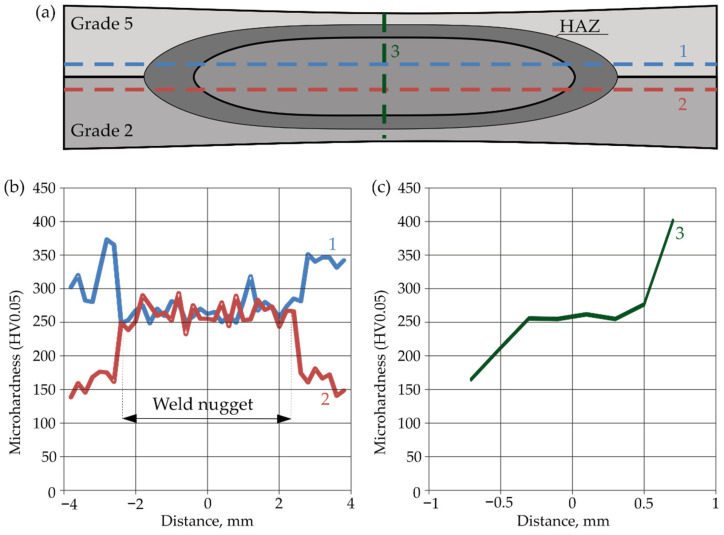
Microhardness of Gr2–Gr5 joints: (**a**) horizontal and vertical measurement lines; (**b**) horizontal hardness distributions and (**c**) vertical hardness distribution.

**Figure 13 materials-16-02038-f013:**
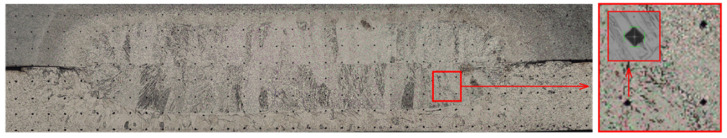
Maps of indentations after hardness measurement.

**Figure 14 materials-16-02038-f014:**
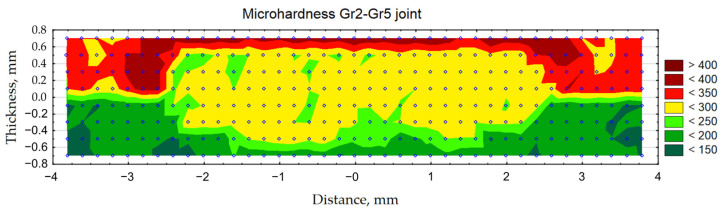
Contour map of the microhardness of the Gr2–Gr5 joint weld.

**Figure 15 materials-16-02038-f015:**
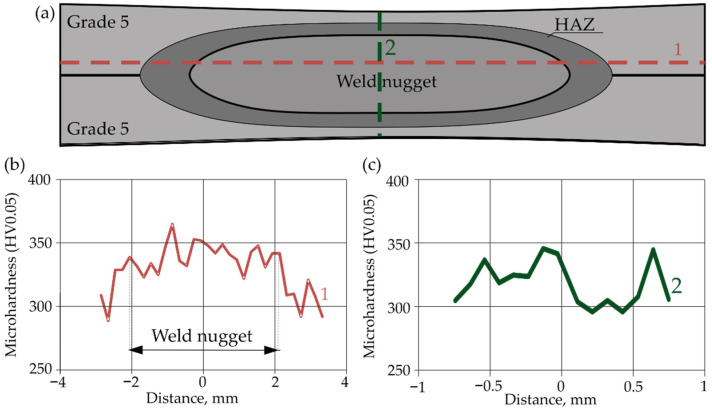
Microhardness Gr5–Gr5 joints: (**a**) horizontal and vertical measurement lines; (**b**) horizontal hardness distributions and (**c**) vertical hardness distribution.

**Figure 16 materials-16-02038-f016:**
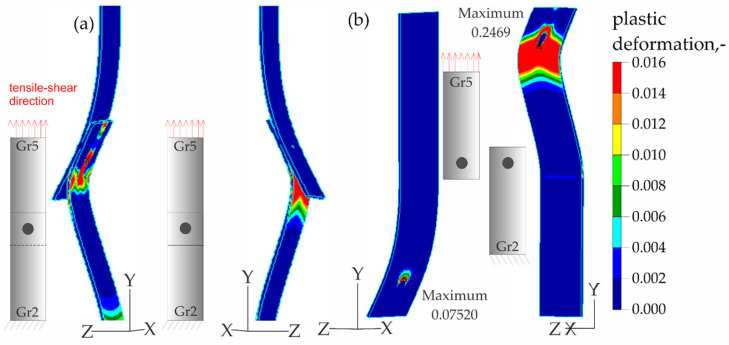
Distribution of plastic deformations in the Gr2–Gr5-1 joints: (**a**) on the surface of the entire joint and (**b**) on the inner surfaces of the joints.

**Figure 17 materials-16-02038-f017:**
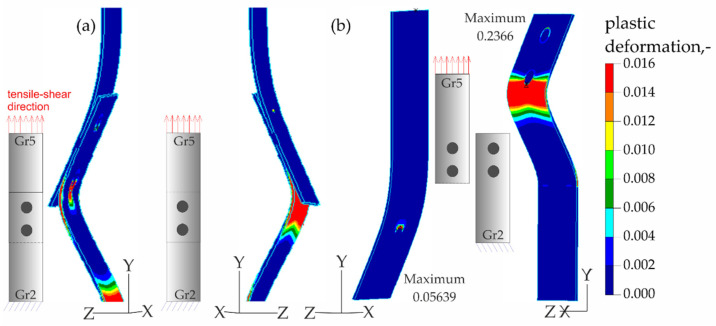
Distribution of plastic deformations in the Gr2–Gr5-2 joints: (**a**) on the surface of the entire joint and (**b**) on the inner surfaces of the joints.

**Figure 18 materials-16-02038-f018:**
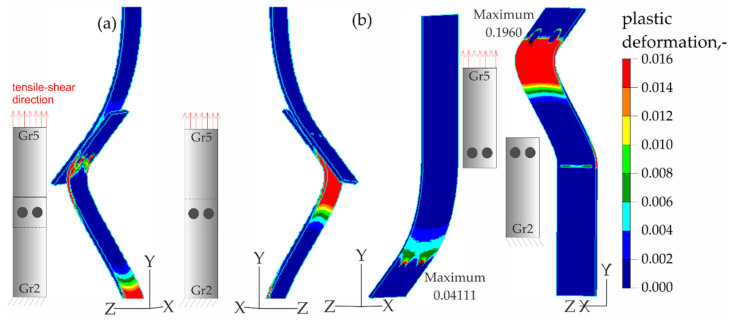
Distribution of plastic deformations in the Gr2–Gr5-3 joints: (**a**) on the surface of the entire joint and (**b**) on the inner surfaces of the joints.

**Figure 19 materials-16-02038-f019:**
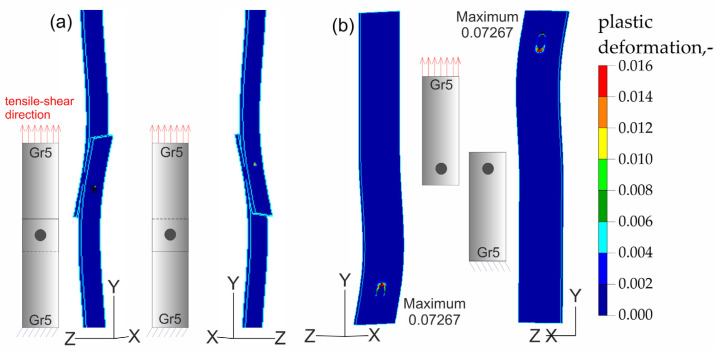
Distribution of plastic deformations in the Gr5–Gr5-1 joints: (**a**) on the surface of the entire joint and (**b**) on the inner surfaces of the joints.

**Figure 20 materials-16-02038-f020:**
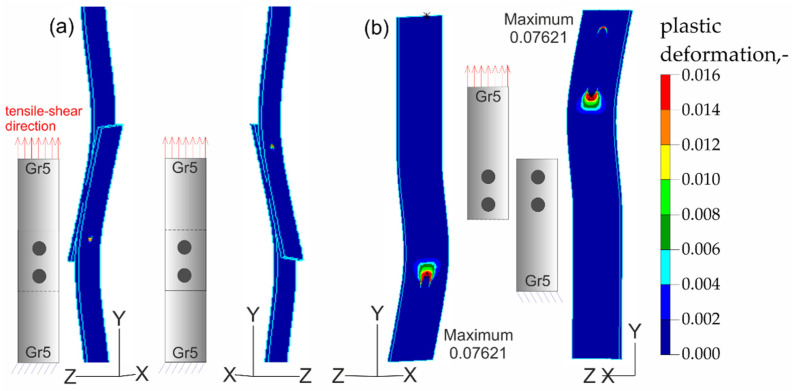
Distribution of plastic deformations in the Gr5–Gr5-2 joints: (**a**) on the surface of the entire joint and (**b**) on the inner surfaces of the joints.

**Figure 21 materials-16-02038-f021:**
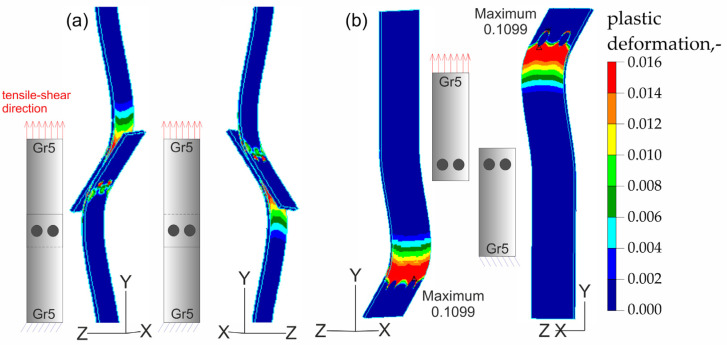
Distribution of plastic deformations in the Gr5–Gr5-3 joints: (**a**) on the surface of the entire joint and (**b**) on the inner surfaces of the joints.

**Figure 22 materials-16-02038-f022:**
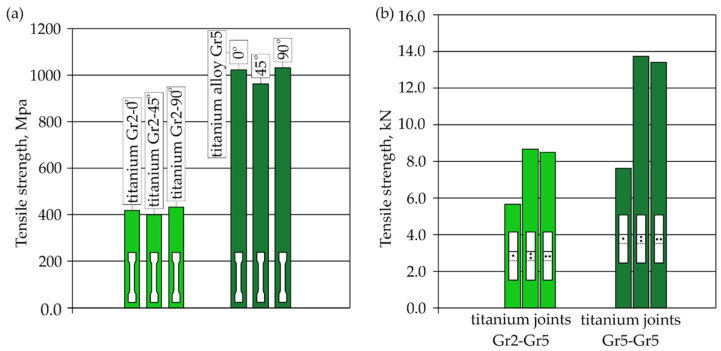
Summary of results: (**a**) the average tensile strength of the analyzed materials depending on the angle at which the sample was taken from the sheet and (**b**) average tensile shear resistance of analyzed RSW joints.

**Figure 23 materials-16-02038-f023:**
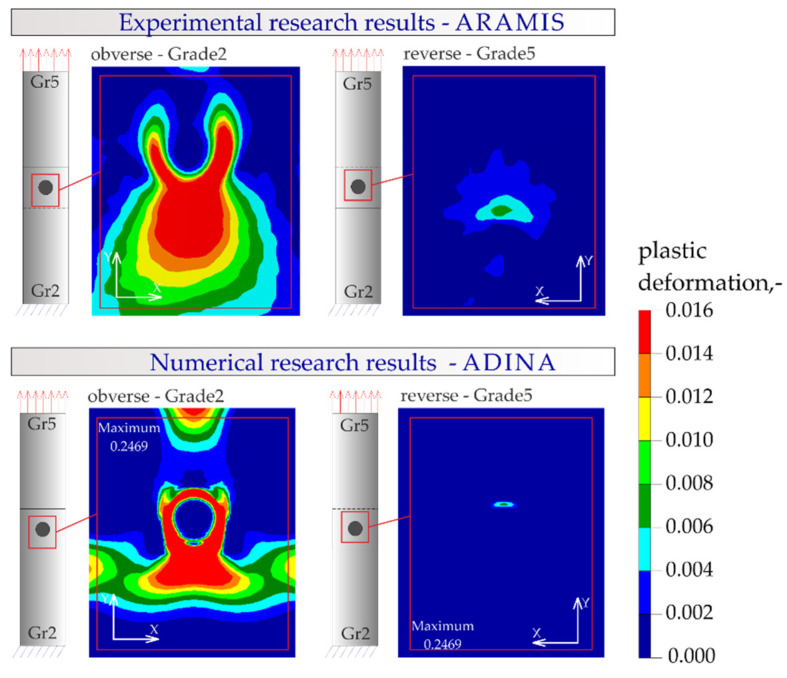
Distribution of plastic deformations in the joints with a single weld—Gr2–Gr5.1.

**Figure 24 materials-16-02038-f024:**
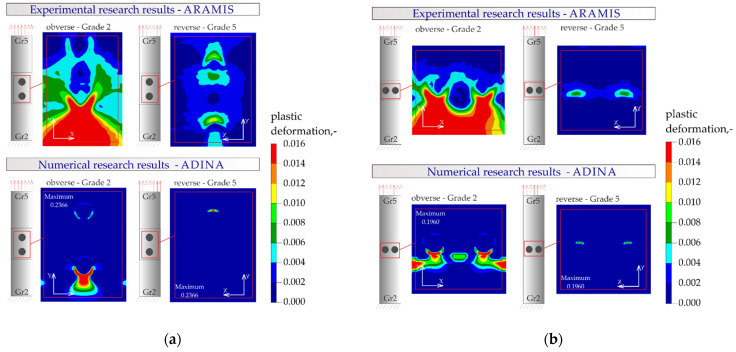
Distribution of plastic deformations: (**a**) in the joints with two welds arranged parallel to the direction of tensile shear—Gr2–Gr5.2 and (**b**) in the joints with two welds arranged perpendicular to the direction of tensile shear—Gr2–Gr5.3.

**Figure 25 materials-16-02038-f025:**
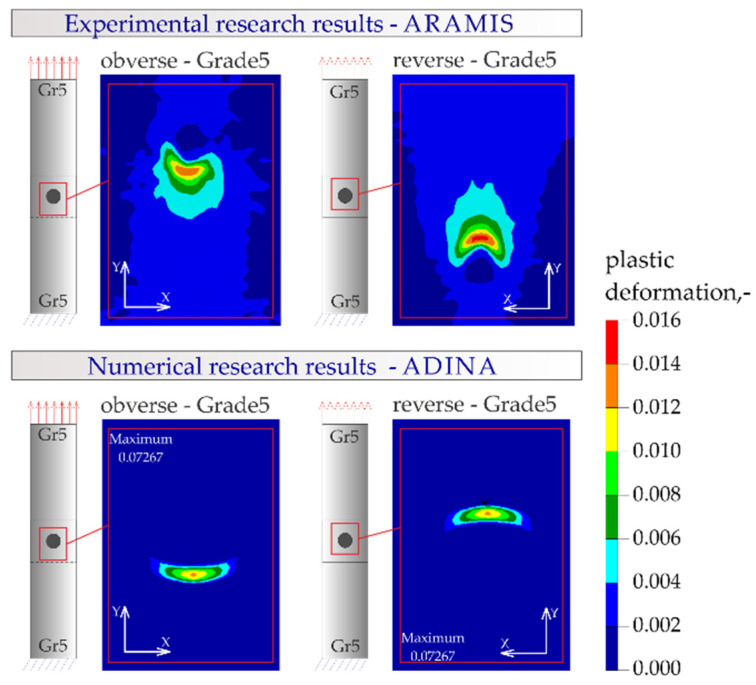
Distribution of plastic deformations in the joints with a single weld—Gr5–Gr5.1.

**Figure 26 materials-16-02038-f026:**
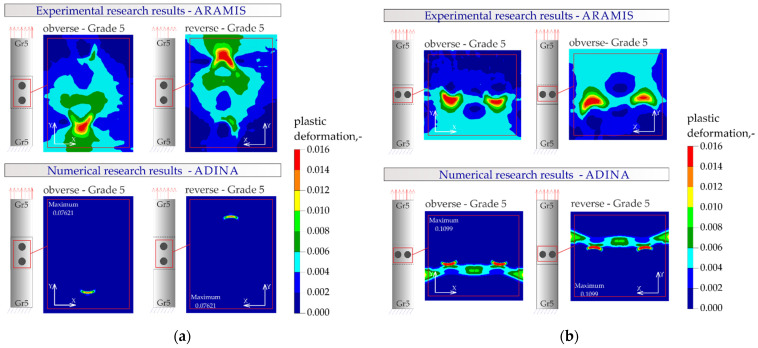
Distribution of plastic deformations: (**a**) in the joints with two welds arranged parallel to the direction of tensile shear—Gr5–Gr5.2 and (**b**) in the joints with two welds arranged perpendicular to the direction of tensile shear—Gr5–Gr5.3.

**Table 1 materials-16-02038-t001:** List of works on resistance spot welding of titanium and titanium alloys.

Paper	Titanium Material Grades	Thickness of Joined Sheets [mm]	Welding Current (kA)	Welding Time (Cycle)	Electrode Force (kN)
[7]	Grade2–Grade2	1.5–1.5	3–7	10–20	3–6
[24]	Grade 5–Grade 5	1.0–1.0	7–11	4–20	2–5
[25]	Grade 3–Grade 4	0.4–0.4	3	4	4
[26]	Grade 5–Grade 5	0.8–0.8	4	5	4
[27]	Grade 2–Grade 5	0.8–0.8	3.35–4.34	9.5	6.2
[27]	Grade 2–Grade 5	0.8–0.8	3.19–3.91	9.5	6.2

**Table 2 materials-16-02038-t002:** Titanium designations according to various standards.

ASTM [28]:	Titanium Grade 2	Titanium Grade 5
UNS:	R50400	R56400
Chemical designations	Ti	Ti-6Al-4V
WNR:	3.7035	3.7165

**Table 3 materials-16-02038-t003:** Chemical composition of technically pure titanium (Grade 2) and Ti6Al4V titanium alloy (wt %).

Material	Component	Al	V	Fe	C	N	H	O	inne	Ti
Grade 2	wt % [29]	-	-	0.3	0.10	0.03	0.015	0.25	≤0.1	rest
Grade 5	wt % [30]	4.5–5.5	3.5–4.5	≤0.3	≤0.1	≤0.05	≤0.015	≤0.2	≤0.1	rest

**Table 4 materials-16-02038-t004:** Welding parameters Gr2–Gr5 and Gr5–Gr5 joints.

Sample Number	Joint Type	Welding Parameters		
		Weld Current (kA)	Welding Time (Cycle)	Electrode Force (kN)
Gr2–Gr5-1.1-5	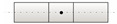	3.75–3.9	9.5	6.2
Gr2–Gr5-2.1-5	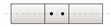	3.35–4.34	9.5	6.2
Gr2–Gr5-3.1-5	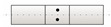	3.87–4.33	9.5	6.2
Gr5–Gr5-1.1-5	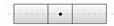	3.19–3.91	9.5	6.2
Gr5–Gr5-2.1-5	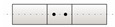	3.39–3.53	9.5	6.2
Gr5–Gr5-3.1-5	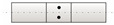	3.48–3.87	9.5	6.2

**Table 5 materials-16-02038-t005:** The total number of finite elements and nodes of the models.

	Joints with Single Weld	Joints with Two Welds—Parallel to the Direction of Tensile Shear	Joints with Two Welds—Perpendicular to the Direction of Tensile Shear
Number of finite elements	5024	5296	4756
Number of nodes	61,468	64,432	57,928

**Table 6 materials-16-02038-t006:** Results from the tensile shear test of Gr2–Gr5 joints.

Sample Number	Thickness [mm]	Joints Type	Force Max [kN]	Average Force [kN]	Displacement at Fmax [mm]	Average Displacement [mm]
Gr2–Gr5-1.1	0.8–0.8	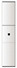	5.72	5.68	1.67	
Gr2–Gr5-1.2	0.8–0.8	5.66	1.91	1.79
Gr2–Gr5-1.3	0.8–0.8	5.66	1.80	
Gr2–Gr5-2.1	0.8–0.8	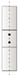	8.65	8.65	5.81	
Gr2–Gr5-2.2	0.8–0.8	8.59	5.92	5.99
Gr2–Gr5-2.3	0.8–0.8	8.70	6.25	
Gr2–Gr5-3.1	0.8–0.8	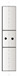	8.49	8.48	8.21	
Gr2–Gr5-3.2	0.8–0.8	8.53	9.01	8.94
Gr2–Gr5-3.3	0.8–0.8	8.41	9.61	

**Table 7 materials-16-02038-t007:** Results from the tensile shear test of Gr5–Gr5 joints.

Sample Number	Thickness [mm]	Joints Type	Force Max [kN]	Average Force [kN]	Displacement at Fmax [mm]	Average Displacement [mm]
Gr5–Gr5-1.1	0.8–0.8	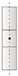	7.89	7.63	1.50	
Gr5–Gr5-1.2	0.8–0.8	7.52	1.44	1.46
Gr5–Gr5-1.3	0.8–0.8	7.47	1.45	
Gr5–Gr5-2.1	0.8–0.8	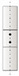	13.72	13.75	2.25	
Gr5–Gr5-2.2	0.8–0.8	13.78	2.16	2.20
Gr5–Gr5-2.3	0.8–0.8	13.85	2.20	
Gr5–Gr5-3.1	0.8–0.8	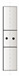	13.81	13.43	2.25	
Gr5–Gr5-3.2	0.8–0.8	13.60	2.22	2.20
Gr5–Gr5-3.3	0.8–0.8	12.88	2.13	

## Data Availability

The data presented in this study are available on request from the corresponding author.

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
