# Peer review of "Numerical and Experimental Analysis of Lap Joints Made of Grade 2 Titanium and Grade 5 Titanium Alloy by Resistance Spot Welding"

_materials, 2023, doi:10.3390/ma16052038_

Round 1

Author Response

Responses to Review 1

I am very much thankful to the reviewers for their deep and thorough review of the manuscript entitled “Numerical and experimental analysis of lap joints made of Grade 2 titanium and Grade 5 titanium alloy by resistance spot welding”. I have revised my present research paper in the light of their useful suggestions and comments. I hope my revision has improved the paper to a level of their satisfaction. My answers to their specific comments/suggestions/queries are as follows.

Abstract

  1. The main idea, reason, and novelty of the work are missing

The main idea of the work was to reaserch the effect of the number and arrangement of welds on the load capacity of a resistance spot welded joint and to verify the distribution of plastic deformations using the numerical method. In the studies carried out so far, the authors have focused on the validation of plastic distributions, methods of destruction of joints with a single weld. The study of the influence of the arrangement and number of joints, the difference in their load capacity, the distribution of plastic deformations is important in the case of analyzing the connections of elements of various structures.

The abstract was supplemented with a sentence containing the main idea and reason for the work:

The influence of the number and arrangement of welds on the load-bearing capacity of the joints and the method of their failure was investigated.line 12 ( line numbers correspond to lines with track changes and comments enabledd)

  1. Authors are suggested to include the key findings in the abstract like:
  2. % change in the load capacity of the joints with the number of welds and their arrangement in the joint.

The abstract was supplemented with the range of % change in load capacity along with the number of welds depending on their arrangement. The exact % values for a specific arrangement in a specific joint type are elaborated in the discussion of results and conclusions.

Gr2-Gr5 joints with two welds, depending on their arrangement, reached from about 149 to 152% of the load capacity of joints with a single weld. The load capacity of Gr5-Gr5 joints with two welds was from about 176 to 180% of the load capacity of joints with a single weld.- line 23

  1. % decrease in average hardness of weld nugget compared to grade 5 titanium alloy.
  2. % increase in average hardness of weld nugget compared to grade 2 titanium.

In the abstract, the percentage value of the hardness of the weld nugget in relation to parent materials Grade2 and Grade 5 has been added

The microhardness test in the Gr2-Gr5 joint showed the average hardness of weld nugget decreased by about 10-23% compared to Grade 5 titanium alloy and increased by about 59-92% compared to Grade 2 titanium. – line 27

Introduction

  1. The overall research gap and need for the described research is missing.

The interest in titanium and titanium alloys (including Grade 5 titanium alloy) was dictated by their wide application in aviation structures. Also, the resistance welding process is a process commonly used in lap joints in the aerospace and automotive industries. A review of the literature pointed to the development, but still few studies on RSW joints of thin-walled titanium elements. The table 1  contains list of tests for specific thicknesses, types of titanium and their welding parameters. On its basis, it was found that further research on RSW titanium joints is an important topic and worth further development. It was also decided to determine the influence of the number and arrangement of welds on the load capacity of the joints.

The introduction was supplemented with the following paragraph:

The choice of Grade 2 titanium and Grade 5 titanium alloy was dictated by their wide application in aircraft structures. The wide application possibilities of these materials generate the need to expand research. Testing the load capacity of titanium joints, and es-pecially the influence of the number and arrangement of welds in the joint, are therefore very important issues in the context of the work and strength of the whole structure..line 171

  1. In line 30, the authors mention “lap joint” but have not elaborated on its significance.

In line 33, an expansion of the meaning "Lap joint" was added. It was indicated that lap joints, where elements overlap each other, are used in large numbers in aircraft structures and in the automotive industry.

  1. In line 63, the authors mention “RSSSW process”. What is it? How RSW process has an

 advantage over it?

The paragraph regarding the RFSSW process (refill friction stir spot welding) has been moved to line 44. Its name has been expanded. It is mentioned and presented because it is an alternative method that can be used e.g. in aircraft structures. However, the RSW and RFSSW process are two different methods of joining materials. The RFSSW process, unlike the RSW process, takes place in the solid state of metals. It consists in local friction heating of the joint by a rotating tool. In the RSW process, the metal melts. The heated metal is mixed under the pressure of the electrodes, creating a permanent connection.

The introduction was supplemented with the following sentence:

An alternative method of joining aluminum sheets is the refill friction stir spot welding (RFSSW) process [4]. It is a process that uses frictional heating of metals in the solid state to join them. It consists in local friction heating of the joint by a rotating tool. In the case of RSW, the joining process takes place in the liquid state of metals.line 44

  1. In line 66, please specify the values of low density and high mechanical strength.

The values of low density and high mechanical strength has been supplemented with references to the literature:

Thanks to their very good properties: low density ranging from 4,43 to 4,85g/cm3 [16] and high mechanical strength ranging from Rm ≈240 MPa for CP1 to about 1750 MPa for the heat treated β titanium alloys [17]line 96

  1. In line 84, the authors mention “the research carried out in [26] was repeated and extended”. The reason for the same is not explained clearly.

The aforementioned, earlier work by the authors [26] concerned the analysis of the bending strength of a composite beam made of Grade 2 titanium and Grade 5 titanium alloy in the RSW technology. RSW welding parameters were selected on the basis of joint strength tests and microstructure analysis. The main attention in this work was devoted to joints with a single weld, for which a preliminary numerical analysis was performed in the Adina System program. In this work, the correctness of the strength tests performed was repeated and verified, and then the focus was on a thorough numerical analysis of all joints. Numerical analysis of joints with two welds arranged parallel and perpendicular to the tensile-shear direction of the joints was carried out. The distribution of plastic strains obtained in experimental studies was verified with the distribution of plastic strains obtained in numerical analysis. This allowed for a more accurate assessment of the influence of the number and arrangement of welds in RSW joints. In order to determine the changes occurring in the structure of the tested materials, additional samples were also made. A more detailed analysis of the microstructure was carried out, and microhardness tests of the tested joints were performed. In order to more accurately illustrate the hardness distribution in the entire joint, a microhardness contour map was made for the Gr2-Gr5 joint. It allowed for accurate depiction of the decrease or increase in hardness in the weld nugget in relation to parent material.

The paragraph has been supplemented with more detailed explanations - line 157

Materials and Methods:

  1. In line 90, the authors mention “Gr2-Gr5” and “Gr5-Gr5”. The reason and explanation behind the selection of these grades for their experiments among other titanium alloys are missing.

The choice of Grade 2 pure titanium and Grade 5 titanium alloy was dictated by their wide application in aircraft structures. Due to the wide range of applications of these materials, it is important to expand the research on how to combine them. The reason and explanation are supplemented in line 171.

  1. The reason behind the selection of sample dimensions and joint variants is not explained in detail.

The dimensions and variant of joints have been adapted to the needs of a local company. In the further perspective of the research, it is planned to optimize the spacing and the number of welds in a specific application.

Results

  1. In line 211, “the joint was completely was broken” should be replaced with “the joint was

 completely broken”.

It has been corrected - line 414

  1. In lines 251 and 254, the authors mention “300÷350 HV” and “140÷170 HV” respectively. What do these mean?

Based on Figures 16 and 17 showing the distribution and contour map of the microhardness of the weld, it was read that the hardness of the base material Grade 5 is from 300 to 350 HV0.05, while the hardness of the base material is from 140 to 170HV0.05 (Vickers method with a 0.05 kgf load test).

„300÷350 HV” and „140÷170 HV” have been changed to from about 300 to 350 HV0,05 and from about 140 to 170HV0,05 line 509 i 512

Discussion

  1. There is no clear and detailed explanation of any of the results mentioned in the manuscript.

The "Discussion" section has been expanded. Detailed percentage relationships of the obtained results and their explanations are presented.

Conclusion

  1. Authors should briefly mention the reason behind the higher tensile shear strength of Grade 5-Grade 5 joints.

The conclusion regarding the joint tensile shear strength has been extended. It was supplemented with the reason for the higher tensile shear strength of the Gr5-Gr5 joints and with the exact percentage relationships.

  1. Authors should mention the % change (increase and decrease) in average hardness and give a

brief explanation of the same.

The % change (increase and decrease) in average hardness to conclusion has been added. A brief explanation is also given(conclusion 5).

Reviewer 2 Report

This study aims to investigate the mechanical and metallurgical soundness of resistance spot welding between two titanium alloy sheets with different weld configurations and then develop a numerical finite element model for the tensile-shear test of the weld to capture the plastic strain distributions and probable failure locations. The idea of the research seems interesting, the manuscript is well organized and the results and discussions are fairly acceptable. However, still many points must be addressed before this study could be accepted for publication in the journal of Materials.

The tense throughout the manuscript is not consistent and must be corrected to be either past or present. The punctuation must be checked and corrected throughout the manuscript.

There are many grammatical errors in the text that must be fixed. Some of them are found below:

-         line 89: The subject of the test were lap joints

-         line 97: a single welds

-         line 169: In the Table 6 summarize the results of

-         line 211: the joint was completely was broken.

A few terms such as “connection” and “destruction” which are used throughout the manuscript, could be replaced with their technical equivalent as “welding, joining” and “failure, fracture”, respectively.

Line 5: The sentence “ Joints made for two combinations of joined titanium sheets …” is difficult to understand and requires rephrasing.

Line 31: the sentence “It is an important process used in industry.” is a very general comment which is valid for the majority of the industrial processes and should be either expanded or removed.

Line 38: in the sentence “ the molten metal solidifies and a uniform join is…”, the word “join” should be replaced with “joint”.

Line 52: The language of the sentence “In the work [9], with ….” is grammatically incorrect and must be corrected.

Line 63: the sentence “An alternative method of joining metal joints is the RFSSW process” seems to be placed improperly and must be displaced into other parts of the introduction. Besides, the authors should expand their explanation of what RFSSW is. Otherwise, they may remove it.

Line 65: this paragraph which is an explanation of titanium could be shortened. The explanations are too long. Also, the references are lacking and must be added to support the information.

Line 77: the three reference numbers should be grouped.

Line 84: The authors mention that this work is an extension of reference [26], so they must explain more regarding the main differences and extensions without a need for the readers to refer to the reference [26].

Section 2:

-         the characteristics of the RSW machine (Brand name, model, …) is missing.

-         the methods utilized for sample preparation for microstructural observation are missing.

-         The norms or standards used for the mechanical tests should be addressed.

-         Microhardness testing details such as load, time, indentor type, … should be stated clearly in the main text as well.

-         The version of the FEM software is missing.

-         Since the model is a 3D one, the number (and size) of the elements through the thickness should be stated.

-         The size of the elements in the most critical region as well as the total number of elements in the model should be stated.

-         The cutting procedure of the samples (which is placed after the numerical modeling,) should be placed where the experimental procedures are being stated.

-         The information in Table 3, Table 4, and Table 5 must be presented at the beginning of section 2, and not at the end of the section.

-         Proper references should be added to Tables 3, 4, and 5.

-         Table 4 and Table 5 should be combined.

Line 111: the language of the sentence is very poor and must be rephrased.

Line 133: the abbreviations (such as FEM) must be defined when used for the first time in the main text.

Section 3

-         Many figures can be combined to reduce the useless total number of the figures: for example figures 6, 7 and 8.

-         Figure 13: it is not clear what the authors intend from figure 13 and how they use this information in the discussions.

-         Section 3.3: some explanation must be provided regarding the microstructures in figures 14 and 15.

-         Line 249: figure 18 is referred to before figure 17 in the text and must be corrected.

-         Figure 16 and 19: the error bars for the data points are missing. How many times were the microhardness tests repeated for each data point?

-         Section 3.5: No procedure is proposed for the validation of the proposed numerical model. The authors could match the output force-displacement curves till the crack initiation point to fulfill this need.

The conclusion must be self-standing and should contain limited information on the scope of the research, and experimental and numerical procedures.

Author Response

Responses to Review 2

I am very much thankful to the reviewers for their deep and thorough review of the manuscript entitled “Numerical and experimental analysis of lap joints made of Grade 2 titanium and Grade 5 titanium alloy by resistance spot welding”. I have revised my present research paper in the light of their useful suggestions and comments. I hope my revision has improved the paper to a level of their satisfaction. My answers to their specific comments/suggestions/queries are as follows.

The tense throughout the manuscript is not consistent and must be corrected to be either past or present. The punctuation must be checked and corrected throughout the manuscript.

The tense and punctuation in the manuscript been corrected

There are many grammatical errors in the text that must be fixed. Some of them are found below:

- line 89: The subject of the test were lap joints – It was corrected – line 178 ( line numbers correspond to lines with track changes and comments enabledd)

- line 97: a single welds - It was corrected – line 212

- line 169: In the Table 6 summarize the results of-  It was corrected – line 341

- line 211: the joint was completely was broken. - It was corrected  – line 414

  1. A few terms such as “connection” and “destruction” which are used throughout the manuscript, could be replaced with their technical equivalent as “welding, joining” and “failure, fracture”, respectively.

Terms :“connection” and “destruction” replaced with their technical equivalent.

Line 5: The sentence “ Joints made for two combinations of joined titanium sheets …” is difficult to understand and requires rephrasing.

The sentence has been rephrased and is now placed on a line 14.

Line 31: the sentence “It is an important process used in industry.” is a very general comment which is valid for the majority of the industrial processes and should be either expanded or removed.

The sentence has been deleted.

Line 38: in the sentence “ the molten metal solidifies and a uniform join is…”, the word “join” should be replaced with “joint”.

The word “join” has been replaced with “joint”.

Line 52: The language of the sentence “In the work [9], with ….” is grammatically incorrect and must be corrected.

The language of the sentence has been corrected and placed in line 83.

Line 63: the sentence “An alternative method of joining metal joints is the RFSSW process” seems to be placed improperly and must be displaced into other parts of the introduction. Besides, the authors should expand their explanation of what RFSSW is. Otherwise, they may remove it.

The paragraph, regarding the RFSSW process (refill friction stir spot welding) has been moved to line 44. Its name and meaning of the process have been expanded. Differences from the RSW process are also listed:

An alternative method of joining aluminum sheets is the refill friction stir spot welding (RFSSW) process [4]. It is a process that uses frictional heating of metals in the solid state to join them. It consists in local friction heating of the joint by a rotating tool. In the case of RSW, the joining process takes place in the liquid state of metals – line 44

Line 65: this paragraph which is an explanation of titanium could be shortened. The explanations are too long. Also, the references are lacking and must be added to support the information.

The paragraph which is an explanation of titanium ( now line 95) has been shortened. Titanium density and strength values have been added with references to the literature.

Line 77: the three reference numbers should be grouped.

The reference numbers have been grouped

Line 84: The authors mention that this work is an extension of reference [26], so they must explain more regarding the main differences and extensions without a need for the readers to refer to the reference [26].

The aforementioned, earlier work by the authors [26] concerned the analysis of the bending strength of a composite beam made of Grade 2 titanium and Grade 5 titanium alloy in the RSW technology. RSW welding parameters were selected on the basis of joint strength tests and microstructure analysis. The main attention in this work was devoted to joints with a single weld, for which a preliminary numerical analysis was performed in the Adina System program. In this work, the correctness of the strength tests performed was repeated and verified, and then the focus was on a thorough numerical analysis of all joints. Numerical analysis of joints with two welds arranged parallel and perpendicular to the tensile-shear direction of the joints was carried out. The distribution of plastic strains obtained in experimental studies was verified with the distribution of plastic strains obtained in numerical analysis. This allowed for a more accurate assessment of the influence of the number and arrangement of welds in RSW joints. In order to determine the changes occurring in the structure of the tested materials, additional samples were also made. A more detailed analysis of the microstructure was carried out, and microhardness tests of the tested joints were performed. In order to more accurately illustrate the hardness distribution in the entire joint, a microhardness contour map was made for the Gr2-Gr5 joint. It allowed for accurate depiction of the decrease or increase in hardness in the weld nugget in relation to parent material.

Section 2:

- the characteristics of the RSW machine (Brand name, model, …) is missing.

RSW machine brand added:

Samples with dimensions of 25x100 mm were cut from Grade 2 and Grade 5 titanium sheets with a thickness of 0.8 mm and then joined by RSW welding on a Soudronic resistance welder – line 185

- the methods utilized for sample preparation for microstructural observation are missing.

The methods utilized for sample preparation for microstructural observation have been added:

The samples were cut perpendicularly to the sheet surface along the axis of the weld, followed ground, polished and etched for metallographic tests. The etching solution consisted of: 2 vol % HN03, 2 vol % HF, 96 vol % H2O – line 222

- The norms or standards used for the mechanical tests should be addressed.

The static tensile test of the analyzed materials were carried out in accordance with the standard PN-EN ISO 6892-1. An appropriate reference was added in the text of the article - line 254. Tensile-shear test of RSW joints was carried out according to the standards of the local enterprise.

- Microhardness testing details such as load, time, indentor type, … should be stated clearly in the main text as well.

Microhardness testing details added in line 227:

The microhardness measurement was performed on the Emco Test Durascan 70 G5 apparatus, Vickers metod with a 0,05 kgf test load. The dwell time was 12s.

- The version of the FEM software is missing.

Numerical analysis was carried out in ADINA System 9.7.2.The version of the FEM software has been added to the text - line 19 and 248

- Since the model is a 3D one, the number (and size) of the elements through the thickness should be stated.

The models had one 27-node element on through the thickness

- The size of the elements in the most critical region as well as the total number of elements in the model should be stated.

The total number of finite elements and nodes of the models has been added in Table 5 - line 267. The size, number of elements and the number of nodes in weld are written on Figure 3.

- The cutting procedure of the samples (which is placed after the numerical modeling,) should be placed where the experimental procedures are being stated.

The cutting procedure of the samples has been transferred in place of specifying experimental procedures

- The information in Table 3, Table 4, and Table 5 must be presented at the beginning of section 2, and not at the end of the section.

The information in Table 3, Table 4, and Table 5 have been transferred at the beginning of section 2

- Proper references should be added to Tables 3, 4, and 5.

Tables 4 and 5 have been combined and appropriate literature references have been added

- Table 4 and Table 5 should be combined.

The tables have been combined – Table 3

Line 111: the language of the sentence is very poor and must be rephrased.

The language of this sentence has been rephrased – line 229

Line 133: the abbreviations (such as FEM) must be defined when used for the first time in the main text.

The abbreviations FEM has been defined in line 20.

Section 3

- Many figures can be combined to reduce the useless total number of the figures: for example figures 6, 7 and 8.

The some of the figures ( including those listed in the example) have been combined

- Figure 13: it is not clear what the authors intend from figure 13 and how they use this information in the discussions.

Figure 13 (currently Figure 9) with the hardening curves of the analyzed materials has been replaced with a diagram showing the hardening curves considered representative.  They were introduced into numerical calculations in the Adina program and allowed for a more accurate description of the materials of the tested joints.

- Section 3.3: some explanation must be provided regarding the microstructures in figures 14 and 15.

Explanations has been added to Figures 14 and 15 (now Figures 10 and 11)

- Line 249: figure 18 is referred to before figure 17 in the text and must be corrected.

The sentence discussing figures 17 and 18 (now 13 and 14) has been corrected – line 506

- Figure 16 and 19: the error bars for the data points are missing. How many times were the microhardness tests repeated for each data point?

For each data point in microhardness profiles pattern, one indent was done, with single microhardness value reading.  Positioning accuracy of indenter was ±0,0035. In addition, for the Grade2-Grade5 joint,  microhardness contour map was made . This allowed for a more accurate assessment of the microhardness distribution over the entire joint.

- Section 3.5: No procedure is proposed for the validation of the proposed numerical model. The authors could match the output force-displacement curves till the crack initiation point to fulfill this need.

The paper focuses on the validation of the proposed numerical model in terms of the distribution of plastic deformations. Experimental an uniaxial tensile test on a tensile testing machine using digital image correlation and tracking (DIC) allowed also to obtain distribution of plastic deformations and obtain maximum strain values. Figures 24-27 present distributions of plastic deformations obtained in both methods. In all analyzed Gr2-Gr5 and Gr5-Gr5 joints, similar distributions were obtained .  The value of deformation in experimental tests corresponded to the values obtained in numerical analysis.

The conclusion must be self-standing and should contain limited information on the scope of the research, and experimental and numerical procedures

The conclusions were supplemented with the percentage dependencies obtained in the research and with short explanations of the results.

Round 2

Reviewer 1 Report

The manuscript looks good to be published. Authors have answered all the comments.

Reviewer 2 Report

The provided responses seem satisfying.